# An Integrated Approach for the Early Detection of Endometrial and Ovarian Cancers (Screenwide Study): Rationale, Study Design and Pilot Study

**DOI:** 10.3390/jpm12071074

**Published:** 2022-06-29

**Authors:** Paula Peremiquel-Trillas, Sonia Paytubi, Beatriz Pelegrina, Jon Frias-Gomez, Álvaro Carmona, José Manuel Martínez, Javier de Francisco, Yolanda Benavente, Marc Barahona, Ferran Briansó, Júlia Canet-Hermida, Víctor Caño, August Vidal, Alba Zanca, Núria Baixeras, Axel Rodríguez, Sergi Fernández-Gonzalez, Núria Dueñas, Laura Càrdenas, Álvaro Aytés, Ilaria Bianchi, Miquel Àngel Pavón, Jaume Reventós, Gabriel Capellà, David Gómez, Mireia Diaz, Jordi Ponce, Joan Brunet, Xavier Matias-Guiu, Francesc Xavier Bosch, Silvia de Sanjosé, Laia Alemany, Marta Pineda, Fátima Marin, Laura Costas

**Affiliations:** 1Cancer Epidemiology Research Programme, Catalan Institute of Oncology, IDIBELL, l’Hospitalet de Llobregat, 08908 Barcelona, Spain; paula.peremiquel@iconcologia.net (P.P.-T.); spaytubic@iconcologia.net (S.P.); bpelegrina@idibell.cat (B.P.); jfrias_ext@iconcologia.net (J.F.-G.); acarmonap@idibell.cat (Á.C.); ybenavente@iconcologia.net (Y.B.); mpavon@iconcologia.net (M.À.P.); dgomez_ext@iconcologia.net (D.G.); mireia@iconcologia.net (M.D.); xbosch.ico@gmail.com (F.X.B.); lalemany@iconcologia.net (L.A.); 2Faculty of Medicine, University of Barcelona, 08036 Barcelona, Spain; 3Consortium for Biomedical Research in Epidemiology and Public Health-CIBERESP, Carlos III Institute of Health, 28029 Madrid, Spain; desanjose.silvia@gmail.com; 4Department of Gynecology, IDIBELL, Hospital Universitari de Bellvitge, Hospitalet de Llobregat, 08908 Barcelona, Spain; jmartinezgar@bellvitgehospital.cat (J.M.M.); mbarahona@bellvitgehospital.cat (M.B.); sergi.sfg@gmail.com (S.F.-G.); jponce@bellvitgehospital.cat (J.P.); 5Department of Anesthesiology, IDIBELL, Hospital Universitari de Bellvitge, Hospitalet de Llobregat, 08908 Barcelona, Spain; javier.defrancisco@bellvitgehospital.cat (J.d.F.); victorc@bellvitgehospital.cat (V.C.); 6Department of Genetics, Microbiology and Statistics, Universitat de Barcelona, 08028 Barcelona, Spain; ferran.brianso@roche.com; 7Roche Diagnostics, Sant Cugat del Vallès, 08174 Barcelona, Spain; 8Hereditary Cancer Group, ONCOBELL Program, Catalan Institute of Oncology, IDIBELL, L’Hospitalet, 08908 Barcelona, Spain; juliaacaanet@gmail.com (J.C.-H.); nduenas@iconcologia.net (N.D.); gcapella@idibell.cat (G.C.); jbrunet@iconcologia.net (J.B.); mpineda@iconcologia.net (M.P.); fmarin@idibell.cat (F.M.); 9Consortium for Biomedical Research in Cancer–CIBERONC, Carlos III Institute of Health, 28029 Madrid, Spain; avidal@bellvitgehospital.cat (A.V.); fjmatiasguiu.lleida.ics@gencat.cat (X.M.-G.); 10Department of Pathology, IDIBELL, Hospitalet de Llobregat, Hospital Universitari de Bellvitge, 08907 Barcelona, Spain; azanca@bellvitgehospital.cat (A.Z.); nbaixeras@bellvitgehospital.cat (N.B.); axelrv93@gmail.com (A.R.); 11Department of Gynecology and Obstetrics, Hospital Universitari Dr. Josep Trueta, 17007 Girona, Spain; lcardenas@iconcologia.net; 12Program against Cancer Therapeutic Resistance (ProCURE), IDIBELL, Hospitalet de Llobregat, 08908 Barcelona, Spain; aaytes@idibell.cat; 13ASSIR Delta, Serveis d’Atenció Primària Delta del Llobregat, Direcció d’Atenció Primària Costa de Ponent, Gerència Territorial Metropolitana Sud, Institut Català de la Salut, 08006 Barcelona, Spain; ilaria-bianchi@hotmail.it; 14Departament de Ciències Bàsiques, Universitat Internacional de Catalunya, 08017 Barcelona, Spain; jaume.reventos@gmail.com; 15Medical Oncology Department, Catalan Institute of Oncology, Doctor Josep Trueta Girona University Hospital, 17007 Girona, Spain; 16Universitat Oberta de Catalunya, 08018 Barcelona, Spain; 17Consultant, National Cancer Institute, Bethesda, MA 20814, USA

**Keywords:** endometrial cancer, ovarian cancer, early detection, pap smears, self-sampling, genomic

## Abstract

Screenwide is a case-control study (2017–2021) including women with incident endometrial and ovarian cancers (EC and OC), BRCA1/2 and MMR pathogenic variant carriers, and age-matched controls from three centers in Spain. Participants completed a personal interview on their sociodemographic factors, occupational exposure, medication, lifestyle, and medical history. We collected biological specimens, including blood samples, self-collected vaginal specimens, cervical pap-brush samples, uterine specimens, and, when available, tumor samples. The planned analyses included evaluation of the potential risk factors for EC/OC; evaluation of molecular biomarkers in minimally invasive samples; evaluation of the cost-effectiveness of molecular tests; and the generation of predictive scores to integrate different epidemiologic, clinical, and molecular factors. Overall, 182 EC, 69 OC, 98 BRCA pathogenic variant carriers, 104 MMR pathogenic variant carriers, and 385 controls were enrolled. The overall participation rate was 85.7%. The pilot study using 61 samples from nine EC cases and four controls showed that genetic variants at the variant allele fraction > 5% found in tumors (*n* = 61 variants across the nine tumors) were detected in paired endometrial aspirates, clinician-collected cervical samples, and vaginal self-samples with detection rates of 90% (55/61), 79% (48/61), and 72% (44/61) by duplex sequencing, respectively. Among the controls, only one somatic mutation was detected in a cervical sample. We enrolled more than 800 women to evaluate new early detection strategies. The preliminary data suggest that our methodological approach could be useful for the early detection of gynecological cancers.

## 1. Introduction

Endometrial cancer (EC) is the most common gynecological tumor in very high human development index regions, and is the second most common globally [1]. Ovarian cancer (OC) is the most lethal gynecological cancer due to diagnosis at an advanced stage [1]. Worldwide, the burden of both tumors is expected to increase in the following years [2,3]. A genetic susceptibility component exists for both tumors: Lynch syndrome (MMR pathogenic variant carriers) is strongly associated with an increased risk of EC, to a lesser extent of OC, while BRCA pathogenic variant carriers have a higher risk of developing OC than the general population [4,5]. Estrogens play a relevant role in EC etiology. Nulliparity, infertility, and age at last birth have been repeatedly associated with EC [6,7,8]. Obesity is also an established risk factor of EC; among women, EC is the most consistently associated with a high body mass index (BMI) [9]. Obesity may increase the risk of EC by a variety of mechanisms, including the conversion of androgens to estrogens via the aromatase activity [10]. OC is also associated with hormonal factors, such as nulliparity, early-onset menarche, late-onset menopause, and the use of hormonal treatments [4,11]. Other factors that interact with sex hormones could potentially modify the risk of EC and OC. Night shift work has been associated with higher concentrations of sex hormones [12], but the role night shift plays in EC and OC is still unknown, with few and discordant results [13,14,15,16,17,18]. Identifying modifiable risk factors for EC and OC is relevant from a public health perspective.

EC arises from malignant transformation of the endometrium via the development of precancerous lesions such as atypical hyperplasia/endometrioid intraepithelial neoplasia (AH/EIN), endometrial glandular dysplasia (EmGD), or serous endometrial intraepithelial carcinoma (SEIC) [19,20]. However, the current paradigm in the genesis of high-grade serous carcinoma, the most common ovarian cancer, consists of a tubal origin of precancerous lesions, such as serous tubal intraepithelial carcinoma (STIC) [21,22]. Theoretically, precancer cells should be molecularly different from the normal endometrium, show a monoclonal growth pattern, and share some but not all features of a malignant endometrium [23]. Around 90% of EC serous cancers show somatic pathogenic variants in *TP53* [24], and these variants have been identified in 43% and 72% of EmGD and SEIC, respectively [25]. Complex AH/EIN shows genetic changes frequently observed in type I endometrial cancer, including microsatellite instability and pathogenic variants in *PTEN*, *PIK3CA*, *KRAS*, and *CTNNB1* [26,27]. Mathematical models suggest that it takes decades for a *TP53* pathogenic variant to develop into a STIC, followed by a shorter span of few years for progression to OC [22].

Currently, no convincing approaches provide the necessary accuracy to be introduced as EC or OC screening tests for the general population. Available strategies to detect EC and OC rely on the presence of symptoms. EC presents abnormal bleeding in up to 90% of cases at diagnosis, allowing for early detection of the disease. Contrary, the broad range of OC symptoms leads to delayed diagnosis in many of the cases, leading to an unfavorable prognosis. Therefore, considerable efforts have been made to implement general population screening to diagnose OC early on, using tools like the transvaginal ultrasound and biomarkers such as serum cancer antigen (CA125), with no significant mortality reductions [28]. Among high-risk populations, such as women with hereditary susceptibility, annual surveillance is recommended until prophylactic surgery, although screening methods are associated with significant pain and distress and include an annual endometrial biopsy. New molecular tests may help refine current diagnostic algorithms among EC symptomatic women by improving the performance and failure rate of histological diagnosis, which currently limits the success of endometrial aspirate-based diagnosis. Accurate non-invasive tests would be especially helpful in screening settings and in high-risk populations, as these suboptimal methods are still used to intensively screen asymptomatic women with a family history of cancer.

The anatomical continuity of the uterine cavity with the cervix represents a unique opportunity to detect signs of disease in the upper genital tract using material from routine cervical pap brush samples (cervical cytology) and other non-invasive sampling methods [29]. Thus, cervical cytology is repeatedly recognized as a potential source of information about gynecological tumors, and molecular approaches (genomic, epigenomic, and proteomic) have been evaluated to detect EC and OC with considerable sensitivity and specificity [30,31,32]. Other minimally invasive methods to detect EC/OC using molecular approaches have been evaluated, such as blood samples, tampons, and vaginal self-samples, with promising results [33,34,35]. These recent findings offer an exciting perspective on the early detection of EC/OC. However, some aspects still need to be assessed to accelerate the implementation of novel technologies in a routine screening or clinical setting. Health economics models, such as cost-effectiveness analyses and budget impact analyses, can help identify the most efficient preventive approaches and inform decision-makers about what screening and early detection strategies may be included in the healthcare plans.

In this context, the Screenwide study was launched in 2017 to evaluate epidemiologic, serological, and genomic factors in the diagnosis of EC and OC for comprehensive primary and secondary prevention of gynecologic cancer in women. Specific objectives include evaluating modifiable risk factors using data from personal questionnaires and from blood biomarkers; evaluating genomic biomarkers in cervical pap samples and vaginal self-collected samples in women with EC/OC and high-risk populations compared with women without cancer; estimating the cost-effectiveness of the introduction of molecular tests at the population level; and generating scores to integrate different epidemiologic, clinical, and molecular factors, to predict the individualized risk of cancer. We show here the enrolment protocols and the methodology used to evaluate genomic biomarkers in minimally invasive samples, and we present the results from the pilot study using 61 samples from nine EC cases and four controls.

## 2. Materials and Methods

Screenwide is a prospective case-control study. Enrolment started in 2017 and finished in 2021. Inclusion criteria included having an intact uterus and, for cases, having an incident diagnosis of EC/OC. Consecutive cases were enrolled during the study period. Gynecologic benign conditions included endometriosis, fibroids, benign cysts, prolapse, and polyps. Hospital controls without gynecologic conditions were enrolled at the anesthetic visits for surgery for conditions, such as ophthalmic or traumatology diseases. High-risk populations without EC/OC were informed about the study at the hereditary cancer clinics and were enrolled during their annual gynecologic check-ups. Exclusion criteria included pregnancy, puerperium (8 weeks), prior treatment with chemotherapy and radiotherapy during the previous 6 months, and communication difficulties that precluded signing informed consent and answering the questionnaire, such as not understanding Spanish or having an intellectual disability. Participants were enrolled in the Bellvitge University Hospital, Josep Trueta University Hospital Catalan Institute of Oncology, and the Sexual and Reproductive Health Care (ASSIR) Delta.

### 2.1. Epidemiological and Clinical Questionnaires

Epidemiological and clinical questionnaires were designed to gather and cover previous existing knowledge on EC and OC. The data collection and entry were regularly monitored for quality control purposes. A structured epidemiological questionnaire was administered by trained personnel in personal interviews. The questionnaire included basic epidemiologic information such as demographic factors, tobacco consumption, lifetime occupational history (including working night shifts), coffee consumption, physical activity, family history of cancer, anthropometric factors, reproductive factors and exogenous hormone use, sun exposure, sleeping habits, and chronotype information.

Clinical data were extracted from the electronic medical records using a predefined form. The information collected included endometrial thickness measured by transvaginal ultrasound, symptomatology (such as abdominal pain, postmenopausal bleeding, or metrorrhagia), tumor stage and grade, tumor type, presence of lymph node invasion, CA-125 levels, treatments received and their duration, and follow-up data on potential relapses.

### 2.2. Biological Samples and Histopathological Examination

Sample collection included blood samples, vaginal samples, cervical pap brush samples, endometrial aspirates, and, when available, tumor samples (Table 1). Gynecological samples were performed in the following order: (1) vaginal sampling, (2) cervical pap brush samples, and (3) endometrial cytologies/aspirates. First, 30 ml of peripheral blood was drawn from participants using an EDTA BD Vacutainer^®^ K2E and SSTTM II Advance BD Vacutainer^®^. The samples^®^ (Becton, Dickinson and Company, Franklin Lakes, NJ, USA). Samples were centrifuged for 15 min at 2500 rpm and the different fractions were aliquoted in whole blood, plasma, cellular fraction for DNA isolation (buffy coat), and serum and were stored at −80 °C. Vaginal samples were collected using the Evalyn brush self-sampling device (Rovers^®^ Medical Devices, The Netherlands) device and were suspended in 5 mL of liquid-based cytology solution (ThinPrep PreservCyt^®^, Hologic, Bedford, MA, USA). Cervical cytologies were collected by the gynecologist using a Cervex brush (Rovers^®^ Medical Devices, The Netherlands) and were suspended in 20 mL of ThinPrep liquid-based solution, as performed in the regular cervical cancer screening program in Catalonia. Once processed, both samples were aliquoted in three vials and were kept at room temperature, as indicated in the manufacturer’s protocol. Some of the vaginal samples and some cytologies were stored at −80 °C for research purposes. Endometrial aspirates were collected by the gynecologist using a pipelle cannula. Half of the sample was formalin-fixed paraffin-embedded for pathology examination, and the remaining fresh sample was frozen at −80 °C. The tumor samples were collected during surgical treatment. Fresh frozen tissues were stored at −80 °C and were formalin-fixed paraffin-embedded and stored at room temperature.

The cervical cytologies, aspirates, and tumor samples were examined by pathologists at the Bellvitge University Hospital and Josep Trueta University Hospital. Pathologic analysis of the hysterectomy and salpingo-oophorectomy specimens were performed in order to (1) confirm the diagnosis of cancer; (2) select different areas for molecular analysis; and (3) check the presence of pathological abnormalities, such as cervical stenosis. Histological evaluation of all of the uterine tissue samples was performed and classified according to the WHO criteria and was staged and graded according to the FIGO classification.

### 2.3. Genomic Biomarkers in Minimally Invasive Sampling Methods—Pilot Study

An analysis of the variants in EC and OC from The Cancer Genome Atlas (TCGA) dataset [36] was performed and, using this along with information from previous literature, we constructed a panel of exonic regions and intron-exon boundaries of 49 genes. In particular, we selected exonic regions from 13 genes that achieved >95% sensitivity to detect EC (*PTEN*, *TP53*, *PIK3CA*, and *ARID1A*, among others), exonic regions from 14 genes that contained frequent point variants (*PPP2R1A*, *RPL22*, *SETD1B*, and *RNF43*, among others), exonic regions from 12 genes including variants among less common histologies (serous cancers; *PIK3R1*, *LZTR1*, *AP4E1*, *ARHGAP35*, among others), and exonic regions from 10 genes selected according to previous literature and our TCGA analyses on ovarian cancer (unpublished; *PAX2*, *AKT1*, *APC*, and *BRAF*, among others). The panel estimated size was 247 kb, and the reference genome used to design the SeqCap EZ probe pool (NimbleGen, Roche) was GRCh38 (hg38) from *Homo sapiens*. Each gene was classified as a tumor suppressor, oncogene, and ambiguous or not driver according to IntOGen [37]. Preparation of the genomic DNA libraries was performed according to the manufacturer’s recommendations, with certain variations to add unique molecular identifiers (UMIs) for duplex sequencing. Target enrichment for next-generation sequencing using SeqCap EZ probes was performed. The deduplication process was performed using a combination of Picard, fgbio, and bwa tools. VarDictJava and Mutect2, in tumor-only mode, were used for variant calling. Variants detected by both callers and with variant allele frequency (VAF) > 0.5% were considered and filtered by quality and functional impact (all non-synonymous and consensus splicing variants were retained). A minimum VAF of 5% was set in aspirates to filter out the low frequency variants of normal tissue. Additional details are provided in the Appendix A.

### 2.4. Statistical and Cost-Effectiveness Analyses

Self-reported data from the epidemiological questionnaires were analyzed to evaluate potential novel risk factors, such as working night shift, early life BMI, and sun exposure. Logistic regression models were adjusted for potential confounders to estimate the odds ratios (OR) and 95% confidence intervals (CI). An algorithm to calculate a predictive score was developed based on regression models, including epidemiologic, clinic, and molecular variables. The contribution of statistically significant variables was determined on a scale of 1 to 10, according to the methodology of Sullivan et al., 2004 [38].

Cost-effectiveness analyses using Markov models were performed to evaluate the best preventive approaches in order to implement different screening and early detection strategies. Different models were designed to assess the impact of the introduction of genomic biomarkers in minimally invasive sampling methods among symptomatic women, women with inherited susceptibility, and the general population. The different strategies were compared using the incremental cost-effectiveness ratio (ICER), expressed as the ratio of the difference in costs (€) between strategies to the difference in effectiveness (QALY). The analyses were performed from the healthcare system perspective. Deterministic and probabilistic sensitivity analyses were carried out to determine the robustness of the results.

### 2.5. Ethical Approval

Screenwide followed all the requirements established by the Ethics Committee for Clinical Research and was approved by the Ethics Committee for Clinical Research from the Bellvitge University Hospital (references: PR128/16 and PR348/19). Participation in the study was voluntary, and all eligible subjects signed an informed consent form after receiving information about the study, before participating in any intervention. The Screenwide study followed the national and international directives on ethics and data protection (Declaration of Helsinki and subsequent amendments; EU Reglament 2016/679) and the Spanish laws on data protection (Organic Law 3/2018; Law 14/2007 biomedical research). The study was registered in the National Register of Biobanks/Collections (C.0004389).

## 3. Results

### 3.1. Overall Enrollment

In total, 838 subjects were enrolled in the study (Table 1), including 251 incident cancer cases (182 EC and 69 OC), 385 controls (119 asymptomatic women in cervical cancer screening programs, 190 hospital controls with benign gynecological pathology, and 76 women attending hospital for non-gynecological diseases frequency-matched to cases by age), and 202 high-risk participants without cancer (98 BRCA and 104 MMR mutation carriers). Participation rates were calculated using the women accepting participation in the numerator and all subjects, including refusals, in the denominator [39]. The response rates among cases were 89.7% for EC and 86.3% for OC. Among the controls, the response rate was 96.7% for asymptomatic women attending cervical cancer screening programs, 80.5% for patients with benign gynecological pathology, and 76.8% for asymptomatic women attending hospital for non-gynecological diseases. Among high-risk populations (BRCA and MMR mutation carriers), the response rate was 85.2% in both groups. Thirty cases (12%) and twenty-one controls (5%) had a previous history of cancer, other than EC or OC. The calculated power was ≥80% to detect associations with a prevalence of exposure among controls between 0.2 and 0.7, and odds ratios of ≥1.8, with the given sample size of 182 cases and 385 controls, or between 0.3 and 0.6 for 182 cases and 266 controls.

In Table 2, we present the main characteristics of the study population. In summary, EC cases were more likely to have a BMI ≥ 30 than the controls (*p* < 0.001), and significant differences were also observed in the distribution of menopausal status and hormonal contraception (*p* < 0.001 and 0.002, respectively, Table 2). Pathologic assessment of cervical pap tests slides was performed. We observed that 26% of slides among EC cases were abnormal [40], while 7% of slides among OC cases were abnormal (contained atypical or malignant cells).

### 3.2. Pilot Study to Detect Somatic Variants in Minimally Invasive Samples

Patients: A total of 61 samples (13 blood samples, 13 vaginal self-collected samples, 13 cervical clinician-collected samples, 13 endometrial aspirates, and 9 tumor samples) were obtained from 13 postmenopausal women (4 controls and 9 sporadic EC cases). The median age was 74 years for cases (range 54–94 years) and 75 for controls (range 58–92). All tumors were histologically confirmed, five cases had endometrioid histology, two cases were serous carcinoma, one was a neuroendocrine tumor (with an endometrioid component), and one was a carcinosarcoma (Appendix A). Four cases were FIGO stage I, four cases were stage III, and one case was stage IV.

Target panel sequencing metrics: The mean raw coverage was 13.477X and final coverage after deduplication and filtering was 868X (Appendix A). No differences were observed between metrics by type of sample (tumor, aspirate biopsy, pap brush sample, and self-sample, *p* > 0.05 for all comparisons).

Surgical specimens: Altogether, 61 variants at VAF > 5% were identified in the tumor samples (Figure 1). All of the tumors harbored at least one known or predicted driver variants of EC (Appendix A). Overall, 17% of variants were identified as known drivers, 52% as predicted drivers, and 31% as passengers. *PTEN* was the gene most mutated in six out of nine EC surgical specimens, followed by *ARID1A* (4/9) and *CDH4* (4/9). Other known driver variants were also found in other genes such as *TP53*, *PIK3CA*, or *CTNNB1*.

Endometrial aspirates: Overall, a total of 58 variants were identified in the endometrial aspirates from EC cases with VAF > 5%, and 55 of them (55/61, 90%) were previously found in paired tumors from surgical specimens (Table 3 and Appendix A). The VAF distributions of the variants in the aspirates were similar to those in the surgical specimens (*p* > 0.05; Figure 2). No variants were detected in the aspirates from the controls (Table 3).

Minimally invasive samples: No variants were detected in one pap brush sample (endometrioid histology) and two vaginal self-samples (endometrioid and serous histologies) from the cases (Table 3, Appendix A). Genetic variants found in tumors at VAF > 5% were detected in paired clinician-collected cervical samples and vaginal self-samples with detection rates of 79% (48/61) and 72% (44/61) through duplex sequencing, respectively (Figure 1). The mean VAF of somatic variants was similar in these minimally invasive samples: 6.4% (range 0.58–40.72%) in pap brush samples and 5.3% (range 0.64–33.19%) in vaginal self-samples, but significantly fewer variants were identified in the aspirates (*p* < 0.001, Figure 2 and Appendix A). In all minimally invasive samples from EC cases where somatic variants could be detected, at least one of them was a predicted or known cancer driver (Appendix A). The mean number of variants in the pap brush samples was 6.8 and 7.2 for stage I and stage > I, respectively, and 6.5 and 7.4, respectively, in the self-samples, although analyses by stage were limited by small sample sizes. Regarding controls, only one variant with VAF > 0.5% was detected in one pap brush sample. This variant (TP53 p.A138V) was not described as a cancer driver, a dominant-negative, nor a loss of function variant. It was not detected in the aspirate or self-sample of the same individual and no (pre)malignant lesions were found in the follow-up after 45 months, suggesting it could correspond to the somatic mutational burden of the normal endometrium [41].

## 4. Discussion

We have presented the protocol of the Screenwide study, which has achieved optimal recruitment based on our target population. The Screenwide study aims to comprehensively evaluate epidemiologic and molecular factors to better understand the epidemiology of EC and OC. We collected detailed epidemiologic data in order to assess new modifiable risk factors. In addition, we will examine sociodemographic, lifestyle, occupational and environmental exposures, as well as EC/OC risk in collaboration with the Epidemiology of Endometrial Cancer Consortium (E2C2). The study has the ambitious mission to characterize molecular markers for the early detection of EC/OC using minimally invasive samples, including vaginal self-collected samples and cervical pap brush samples. The preliminary data suggest that our methodological approach could be useful to evaluate new early detection strategies for gynecological cancers. Finally, our approach will integrate different epidemiologic, clinical, and molecular factors to generate a predictive score to assess individual risk of EC/OC, with the aim to translate the generated epidemiological knowledge into clinical practice. A cost-effectiveness evaluation is expected to identify scenarios that better contribute to the sustainability of healthcare systems with the introduction of these novel technologies.

EC and OC are expected to rise worldwide in the following decades and, currently, neither screening nor efficient early diagnosis approaches exist. Recent developments on non-invasive sampling methods using genomic, epigenomic, and proteomic approaches offer promising prospects for assessing this issue [29]. Vaginal self-sampling is being implemented in many settings to increase cervical cancer screening coverage and is offered as an alternative screening approach in some regions [42]. The evaluation of EC/OC markers in these specimens could provide women with a noninvasive screening option, thus avoiding pain and discomfort, as well as decreasing medical complications. Our approach described in the pilot study is useful to detect somatic variants in these gynecological minimally invasive samples at a high sensitivity. We have observed a few variants in minimally invasive samples that were not present in the tumor samples. This may be due, in part, to the mutational burden of normal endometrium [41]. Tumor heterogeneity could also contribute to this issue, as it hampers genetic characterization when a single tumor biopsy is analyzed, and genetic analyses of uterine aspirates have been shown to capture this heterogeneity [43]. However, we did not always observe concordant variants in aspirates and minimally invasive samples that were not found in tumors, suggesting that our findings might not be due to this phenomenon. We used duplex sequencing, as it has been described as the most reliable method to detect ultra-low frequency variants, and the theoretical error rate has been estimated to be around 1 × 10^−7^ [44]. We enriched our sample for non-endometrioid histologies (four out of nine cases, 44% rather than 10–15%) in order to ensure that we would also detect rarer histologies, which are commonly associated with a poorer prognosis. A larger sample size is required to validate these results. In this regard, we are currently sequencing more than 550 additional samples from EC, OC cases, controls, and high-risk populations, which will be split in a training (70% of the sample) and validation dataset (30% of the sample). This sample size provides power (99%) to detect ≥30% difference in the prevalence of variants. The detection of OC would probably need further adjustments of the bioinformatics pipeline, given the expected lower VAF in endometrial aspirates and minimally invasive samples compared with EC samples. The rest of the collected samples will be used to evaluate other molecular techniques (including other genomic approaches, proteomics, and epigenomics), according to the obtained results, and in collaboration with other partners. As this is an exciting and emerging field of research—novel hypotheses and further opportunities for collaboration are expected.

Because of the retrospective nature of case-control studies, we should consider that certain information obtained may be a result of the disease and not the cause of it. The variables collected in the epidemiological questionnaire are mostly self-reported, and participants may have a recall bias, which could lead to misclassification. The inclusion of controls from higher socioeconomic levels may lead to selection bias. This is especially relevant for controls from cancer screening programs. These controls are expected to contribute mostly to technological development and will be potentially excluded from etiologic factors analyses. In order to control for these possible biases, several quantitative analyses of the selection and misclassification biases of the exposure are planned with plausible parameters under a probabilistic approach. The molecular characterization of tumors allows for defining the molecular phenotype and for evaluating the specific risk factors for each molecular subtype. The available sample size allows for an adequate characterization of genomic biomarkers and permits collaboration with different related consortia, although it might be insufficient to evaluate certain associations, especially in the case of OC.

## 5. Conclusions

We enrolled more than 800 women to evaluate new early detection strategies. The preliminary data suggest that our methodological approach could be useful for the early detection of gynecological cancers. The Screenwide study aims to provide new evidence regarding personalized gynecologic cancer care by offering non-invasive early detection methods for women.

## Figures and Tables

**Figure 1 jpm-12-01074-f001:**
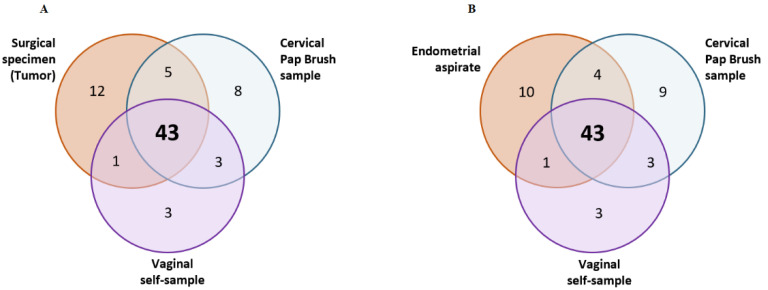
Venn diagrams showing concordance between number of variants detected in the tumor (**A**) or in the endometrial aspirate (**B**) and the minimally-invasive samples.

**Figure 2 jpm-12-01074-f002:**
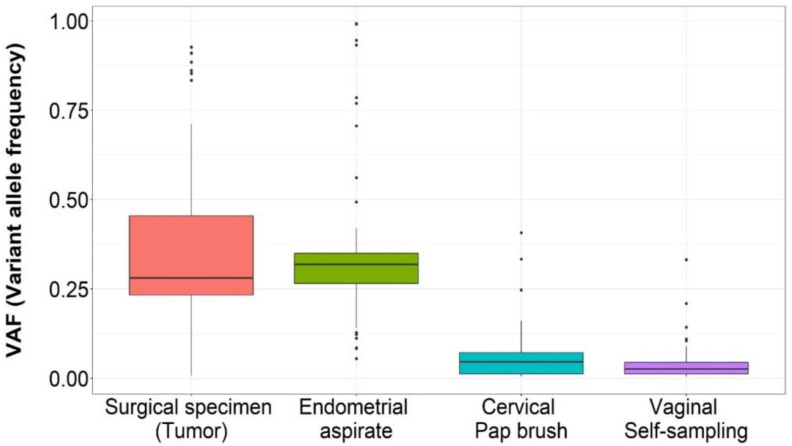
Boxplots for variant allele frequencies by sample type.

**Table 1 jpm-12-01074-t001:** Number of participants, epidemiologic questionnaires, and biologic samples.

	Participants	Epidemiologic Questionnaire	Blood	Vaginal Self-Samples	Cervical Pap-Brush Samples	Endometrial Aspirates	Tumour Samples
Endometrial cancer ^1^	182	180	174	175	168	161	165
Ovarian cancer	69	69	67	66	62	56	52
MMR pathogenic variant carriers ^2^	104	103	1	104	102	90	NA
BRCA pathogenic variant carriers ^3^	98	98	0	98	98	0	NA
Healthy women attending CC screening	119	119	33	119	118	0	NA
Controls with gynaecologic benign conditions	190	146	117	176	151	105	NA
Hospital controls (non-gynaecologic)	76	72	76	48	0	0	NA
TOTAL	838	787	468	786	699	412	217

NA = not applicable; CC = cervical cancer. ^1^ Includes two synchronic (endometrial and ovarian) cancer cases. The table also includes 154 EC cases in which a Lynch syndrome tumor screening was performed with immunohistochemistry of mismatch repair system proteins (MMR). Among these EC cases, 119 were MMR proficient (MMRp) and 35 were MMR deficient (MMRd) cases. MMRd patients included four cases of confirmed Lynch syndrome, 14 presumed sporadic (MLH1 methylated), and 17 under evaluation. ^2^ Women without EC/OC included five women with premalignant lesions (three cases of complex atypical hyperplasia; one case of complex hyperplasia without atypia, and one case of simple atypical hyperplasia), 17 women with a history of colorectal cancer, 3 women with a previous breast cancer diagnosis, and 3 women with other cancer types. ^3^ Women without EC/OC included 32 women with a history of breast cancer.

**Table 2 jpm-12-01074-t002:** The main characteristics of the population of the Screenwide study.

	High Risk Populations ^1^	Controls ^2^	EC ^3^	OC	*p*-Values ^4^
	*n* (%)	*n* (%)	*n* (%)	*n* (%)	
**Participants**	202	385	182	69	
**Epidemiologic questionnaire**	201 (99.5)	337 (87.5)	180 (98.9)	69 (100.0)	
**Age**					0.740/0.124
<60	188 (93.1)	165 (42.9)	51 (28.0)	26 (37.7)	
60–69	12 (5.9)	108 (28.1)	61 (33.5)	22 (31.9)	
≥70	2 (1.0)	112 (29.1)	70 (38.5)	21 (30.4)	
**Education ^5^**					0.426/0.997
High School or below	63 (31.3)	240 (71.2)	131 (72.8)	52 (75.4)	
Some college/associate	64 (31.8)	70 (20.8)	29 (16.1)	12 (17.4)	
College or above	74 (36.8)	27 (8.0)	20 (11.1)	5 (7.2)	
**BMI ^5^**					<0.001/0.500
<18.5	10 (5.0)	4 (1.2)	1 (0.6)	3 (4.3)	
18.5–24.99	110 (54.7)	99 (29.4)	26 (14.4)	21 (30.4)	
25–29.99	46 (22.9)	131 (38.9)	52 (28.9)	24 (34.8)	
≥30	28 (13.9)	89 (26.4)	95 (52.8)	20 (29.0)	
**Previous history of cancer (other than EC or OC)**					0.346/0.791
Yes	56 (27.7)	47 (12.2)	29 (15.9)	8 (11.6)	
No	146 (57.2)	338 (87.8)	153 (84.1)	61 (88.4)	
**Menopausal status ^5^**					0.001/0.633
Premenopausal	151 (75.1)	63 (18.7)	7 (3.9)	13 (18.8)	
Perimenopausal	14 (7.0)	29 (8.6)	12 (6.7)	2 (2.9)	
Postmenopausal	36 (17.9)	245 (72.7)	161 (89.4)	54 (78.3)	
**Parity ^5^**					0.792/0.122
Nulliparous	77 (38.3)	36 (10.7)	26 (14.4)	10 (14.5)	
1	43 (21.4)	70 (20.8)	28 (15.6)	20 (29.0)	
≥2	81 (40.3)	229 (68.0)	126 (70.0)	39 (56.5)	
**Hormonal contraception^5^**					0.002/0.483
Never	53 (26.4)	114 (33.8)	107 (59.4)	34 (49.3)	
Ever	148 (73.6)	221 (65.6)	72 (40.0)	35 (50.7)	
**Tobacco consumption ^5^**					0.651/0.794
Never	81 (40.3)	193 (57.3)	126 (70.0)	48 (69.6)	
Ever	120 (59.7)	144 (42.7)	54 (30.0)	21 (30.4)	

BMI = body mass index. ^1^ Includes BRCA and MMR pathogenic variants carriers. ^2^ Includes women attending cervical cancer screening programs, patients with a benign gynecological pathology, and non-gynecological hospital controls. ^3^ Includes two synchronic (endometrial and ovarian) cancer cases. ^4^
*p*-value for controls and EC, and OC, respectively, excluding women attending cervical cancer screening programs. ^5^ Data obtained from epidemiological questionnaires, numbers do not always add up due to missing data. Missing data are <5% for all variables.

**Table 3 jpm-12-01074-t003:** Number of somatic variants found in gynaecological samples compared with variants identified in tumor samples in the pilot study.

Patient ID	Case-Control Status	Tumor ^1^	Endometrial Aspirates ^2^		Cervical Pap Brush Samples ^3^		Vaginal Self-Samples ^3^	
		Total Nº variants	Total Nº variants		Total Nº variants		Total Nº variants	
S110444	Control	NA	0		0		0	
S110449	Control	NA	0		1		0	
S110565	Control	NA	0		0		0	
S110574	Control	NA	0		0		0	
		Total Nº variants	Total Nº variants	Nº variants also identified in tumour	Total Nº variants	Nº variants also identified in tumour	Total Nº variants	Nº variants also identified in tumour
S110036	Case	4	4	4	4	3	2	2
S110081	Case	25	20	20	24	21	19	19
S110124	Case	4	4	4	0	0	0	0
S110424	Case	3	4	3	3	3	2	2
S110435	Case	5	5	5	5	5	5	5
S110451	Case	3	3	3	1	1	0	0
S110474	Case	11	12	10	14	10	16	10
S110475	Case	5	5	5	6	4	5	5
S110501	Case	1	1	1	1	1	1	1
Total		61	58	55	58	48	50	44
% (95% CI)			95 (86–98) ^5^	90 (80–95) ^4^	84 (73–92) ^5^	79 (67–87) ^4^	90 (79–96) ^5^	72 (60–82) ^4^

Nº = Number of. NA = not applicable. For each type of sample, other than the tumor samples, two columns are specified for the following cases: the total number of variants in that sample and the number of variants that are jointly identified in the tumor. ^1^ Nº of variants identified in tumor surgical specimens (cases). ^2^ VAF > 5%. ^3^ VAF > 0.5%. ^4^ Nº variants also identified in the tumor/Nº variants in the tumor. ^5^ Nº variants also identified in the tumor/Nº variants in the corresponding sample.

## Data Availability

Data can be found in Appendix A.

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
