# Peer review of "An Integrated Approach for the Early Detection of Endometrial and Ovarian Cancers (Screenwide Study): Rationale, Study Design and Pilot Study"

_jpm, 2022, doi:10.3390/jpm12071074_

Round 1

Reviewer 1 Report

The authors present their Screenwide case-control research study, and the initial results of a pilot genomic study to detect somatic mutations in minimally invasive samples.

There are some grammatical changes required to address the English language and style. For example, the title ".... to early detect endometrial and ovarian cancers" should probably be "for the early detection of ...". There are multiple instances of this type of language throughout the manuscript.

Commentary about what proportion of the overall EC/OC population have been captured by Screenwide in this time period would be useful  - not just the response rate amongst approached patients/participants.

Commentary about the likelihood of BRCA and MMR mutation positive patients amongst the cases is missing. Presumably this data will be captured and that is the reason for inclusion of the high-risk control group?

A break down of the detected variants in the pilot study by tumor stage (in the instance of the cases) would lend weight to the use of minimally invasive samples as a screening option. What was the pattern of variant detection in the  Stage I case versus the Stage III or IV cases? Sub-analyses by histological type would be valuable in a larger series.

What was the suggested cause/action taken with regards to the TP53 mutation identified in the control? Is this evidence of a pre-cursor to a malignant lesion?Commentary by the authors would be appropriate.

Presumably the variants detailed in Table 3 are the same per individual? ie total found in tumor (n=4), so the 4 variants identified in the endometrial aspirate are the same 4? some clarification would be useful.

Reviewer 2 Report

In the study titled “An integrated approach to early detect endometrial and ovarian cancers (Screenwide Study): Rationale, study design and pilot study” the authors conducted a case-control study including patients with endometrial and ovarian cancer and also proposed a large-scale study which will be done in future based on the results of this study. However, the pilot study conducted with samples from nine cases is valuable and could be useful in early detection of gynecological cancers. A larger study is warranted as proposed, to establish this technique as standard tool for early detection of endometrial and ovarian cancers. The manuscript could be accepted after following major revisions.

Major Revisions

1.       Molecular clinicopathological evidence demonstrated cancer development from genetically altered cells is a step wise progression including hyperplasia, dysplasia, carcinogenesis in situ and malignant tumors, along with a complex process of sub clone. The authors should involve all the precancerous lesions in to their study if possible.

2.       Previous studies have demonstrated that cancer development derived by driver mutations accompanied by accumulation of molecular events, not only mutations but also CNV, chromosome instability and MSI etc. therefore these molecular events analysis should also be included in this study.

3.       In the pilot study, the NGS sequencing only covered the exons, what about the splicing sites? The splicing sites should be covered as well as the splicing site variants always have a pathogenic effect.

4.       Out of 202 high risk participants, 104 were mentioned as lynch syndrome individuals. Authors should clarify whether these 104 individuals were non-endometrial cancer patients having other tumors or healthy individuals that carry lynch syndrome mutations, in case of later situation, they should be mentioned as MMR gene mutation carriers.

5.       In table 3, authors reported that less mutations were detected in tumor tissue compared to that in cervical pap brush samples. Please explain if these patients are HPV (human papillomavirus) positive or not, as HPV virus also causes several mutations that ultimately lead to tumor development, hence the detected mutations might be result of two disorders collectively (i.e. HPV and endometrial tumor).

6.       As the pilot study not focuses specifically on the pathogenic mutations, the term “mutations” should be replaced with “variants” in the manuscript.

7.       In the “Introduction” section, the repetition of references inside a single paragraph can be avoided by mentioning the reference once at the end of the paragraph or by adding some more recent references. 

Reviewer 3 Report

(1)   The publication of this Screenwide study had better be deferred until the completion of all data analysis.

(2)   The “high-risk population”, “EC population” and “OC population” have different characteristics. Is it possible to arrange individualized control groups for the “high-risk population”, “EC population” and “OC population”?

(3)  In Screenwide study, the blood, vaginal samples, cervical Pap brush samples, endometrial aspirates were sampled only once. However, in the “high-risk population”, the results might change gradually year by year. Should the “high-risk population” be sampled serially along their lifetime? 

(4)  In lines 70-72, “Nulliparity, infertility, early age at menarche, and use of exogenous hormones, such as oral contraceptives and postmenopausal hormone therapy, have been repeatedly associated with EC [4,5].” However, in fact, with the addition of progestin, combinational oral contraceptives and combinational postmenopausal hormone therapy are unlikely to increase the risk of endometrial cancer.

Round 2

Reviewer 1 Report

Thank you to the authors for addressing the previous feedback,

Reviewer 3 Report

May the funding for your longitudinal study be approved soon.